# Obese Patients Experience More Severe CSA than Non-Obese Patients

**DOI:** 10.3390/ijerph19031289

**Published:** 2022-01-24

**Authors:** Yao-Ching Huang, Shi-Hao Huang, Ren-Jei Chung, Bing-Long Wang, Chi-Hsiang Chung, Wu-Chien Chien, Chien-An Sun, Pi-Ching Yu, Chieh-Hua Lu

**Affiliations:** 1Department of Chemical Engineering and Biotechnology, National Taipei University of Technology (Taipei Tech), Taipei 10608, Taiwan; ph870059@gmail.com (Y.-C.H.); hklu2361@gmail.com (S.-H.H.); rjchung@ntut.edu.tw (R.-J.C.); 2Department of Medical Research, Tri-Service General Hospital, Taipei 11490, Taiwan; g694810042@gmail.com; 3School of Public Health, National Defense Medical Center, Taipei 11490, Taiwan; billwang1203@gmail.com; 4Graduate Institute of Life Sciences, National Defense Medical Center, Taipei 11490, Taiwan; 5Taiwanese Injury Prevention and Safety Promotion Association (TIPSPA), Taipei 11490, Taiwan; 6Department of Public Health, College of Medicine, Fu-Jen Catholic University, New Taipei City 242062, Taiwan; 040866@mail.fju.edu.tw; 7Big Data Center, College of Medicine, Fu-Jen Catholic University, New Taipei City 242062, Taiwan; 8Graduate Institute of Medicine, National Defense Medical Center, Taipei 11490, Taiwan; yupichin1003@gmail.com; 9Cardiovascular Intersive Care Unit, Department of Critical Care Medicine, Far-Eastern Memorial Hospital, New Taipei City 10602, Taiwan; 10Division of Endocrinology and Metabolism, Department of Internal Medicine, Tri-Service General Hospital, School of Medicine, National Defense Medical Center, Taipei 11490, Taiwan

**Keywords:** central sleep apnea, obesity, National Health Insurance Research Database, case-control study

## Abstract

Objective: To investigate whether central sleep apnea (CSA) is associated with an increased risk of obesity. Materials and methods: From 1 January 2000 to 31 December 2015, we screened 24,363 obese patients from the 2005 longitudinal health insurance database, which is part of the Taiwan National Health Insurance Research Database. From the same database, 97,452 non-obese patients were also screened out. Age, gender, and index dates were matched. Multiple logistic regression was used to analyze the previous exposure risk of obese and CSA patients. A *p*-value of <0.05 was considered significant. Results: Obese patients were more likely to be exposed to CSA than non-obese patients would (AOR = 2.234, 95% CI = 1.483–4.380, *p* < 0.001). In addition, the closeness of the exposure time to the index time is positively correlated with the severity of obesity and has a dose–response effect (CSA exposure < 1 year, AOR = 2.386; CSA exposure ≥ 1 year and <5 years, AOR = 1.725; CSA exposure time ≥ 5 years, AOR = 1.422). The CSA exposure time of obese patients was 1.693 times that of non-obese patients. Longer exposure time is associated with more severe obesity and has a dose-response effect (CSA exposure < 1 year, AOR = 1.420; CSA exposure ≥ 1 year and <5 years, AOR = 2.240; CSA ≥ 5 years, AOR = 2.863). Conclusions: In this case-control study, patients with CSA had a significantly increased risk of obesity. Long-term exposure to CSA and obesity is more likely and has a dose-response effect.

## 1. Introduction

Overweight and obesity are defined as abnormal or excessive fat accumulation that presents a risk to health. A body mass index (BMI) over 25 is considered overweight, and over 30 as obese [1]. Obesity is a serious disease that causes serious health damage. Obese adults have an increased risk of death due to various acute and chronic diseases (including hypertension, dyslipidemia, coronary heart disease, diabetes, gallbladder disease, gout, arthritis, and respiratory diseases) [2]. With the current growth rate, by 2025, the global obesity rate for men and women will reach 18% and 21%, respectively [3].

According to the latest survey report of the Ministry of Health and Welfare, the obesity rate of adults (≥18 years of age) has increased from 38% in 2009 to 43.9% in 2018 [4]. The risk of obese people suffering from diabetes mellitus (DM), metabolic syndrome, and dyslipidemia is more than three times that of people with normal weight, while the risk of suffering from hypertension, cardiovascular disease (CVD), knee arthritis, and gout is twice as much [5].

A prevalence of obstructive sleep apnea syndrome (OSAS) between 19% and 61% has been found in obese children and adolescents [6]. Although several studies have identified obesity as an independent risk factor for polysomnography (PSG)-confirmed OSAS, other reports have not reproduced this relationship [7,8]. It has been suggested that OSAS is independently associated with the metabolic syndrome and its components and thus OSAS is potentially an additional risk factor for the development of cardiovascular morbidity in obese children [9,10].

Adenotonsillectomy as a treatment intervention for OSAS in childhood is frequently accompanied by accelerated weight gain, even in children who are overweight preoperatively [11]. A further increase in adiposity in overweight children places them at additional risk of OSAS and the adverse consequences of obesity [12]. A recent longitudinal population-based study indicated that cases of prepubertal SDB tend to resolve naturally during the transition to adolescence, but obesity was a risk factor for SDB in adolescent years [13,14]. As reported, sleep fragmentation and chronic inflammation status could lead to neurocognitive disorders with poor cognitive functions and memory up to degenerative diseases as Alzheimer’s disease (AD) or Parkinson’s [15]. The immediate deleterious effect of OSA on cognition, especially on executive function and attention, may contribute to a worsening of the AD clinical presentation [16].

Central sleep apnea (CSA) is characterized by insufficient respiratory function during sleep leading to repeated hypoventilation and impaired gas exchange [17]. Similar to obstructive sleep apnea, the clinical symptoms of CSA include fatigue, daytime sleepiness, and nocturia [18]. In addition, the proportion of CSA that stops snoring is relatively low, and the degree of lethargy is not as serious as obstructive sleep apnea [19]. CSA, similar to OSA, is associated with important complications, including frequent nocturnal awakenings, excessive daytime sleepiness, and an increased risk of cardiovascular disease. Repeated awakenings of brain waves result in sleep fragmentation, poor concentration and memory, emotional instability, daytime sleepiness, and decreased quality of life [20].

However, CSA may be a key mediating factor linking sleep disorders with chronic diseases of all ages (including CVD and DM) [21,22]. Understanding this connection may contribute to effective therapeutic interventions for sleep disorders and obesity [23,24]. At present, longitudinal observational studies on the relationship between sleep disorders and obesity are limited. Therefore, we hypothesize that central sleep apnea is related to obesity. We used the National Health Insurance Research Database (NHIRD) of the Ministry of Health and Welfare to investigate whether CSA increases the risk of subsequent obesity.

## 2. Materials and Methods

### 2.1. Data Source

Taiwan’s National Health Insurance launched a single payment system on 1 March 1995. As of 2017, 99.9% of Taiwan’s population participated in the program. The data for this study comes from the 2005 Longitudinal National Health Insurance Database (LHID2005), which is part of the NHIRD, and a randomly selected 2,000,000 people from the entire population. The National Institutes of Health encrypted all personal information before they released LHID2005 to protect the privacy of the patients. In LHID2005, the disease diagnosis code is based on the “International Classification of Diseases, Ninth Revision, Clinical Modification” (ICD-9-CM) standard [23]. The research design flow chart of this study (case-control study) is shown in Figure 1. All the procedures performed in this research involving human participants comply with the ethical standards of the institution and/or the National Research Council and comply with the 1964 Declaration of Helsinki and its subsequent amendments or similar ethical standards. All methods were carried out following the relevant guidelines and regulations. The informed consent of all subjects was obtained; if the subject was under 18 years of age, the informed consent of the parents and/or legal guardians was obtained. The Ethical Review Board of the General Hospital of the National Defense Medical Center (TSGHIRB B-109-39) approved this study.

### 2.2. Case Group and Control Group

The patients diagnosed as obese (ICD-9-CM code 278) constitute the obese case group. The control group consisted of patients who were not obese. The patients in the case group and the control group were matched with index date, gender, and age in a ratio of 1:4.

### 2.3. Definition

CSA, obesity, and comorbidities: The risk factor discussed in this study is CSA, which is defined based on at least three outpatient diagnoses from 2000 to 2015. The ICD-9 code 780.5 (sleep disorder) was used for identification as follows: 780.50 (sleep disorder, not specified); 780.52 (other insomnia, not specified); 780.51, 780.53, and 780.57 (sleep apnea syndrome); 307.4 (specified nonorganic sleep disorders); 780.54 (other drowsiness, not specified); 780.55 (24 h sleep-wake cycle interruption, not specified); 780.56 (dysfunction during sleep phase and waking from sleep); 780.58 (sleep-related movement disorders, unspecified); and 780.59 (sleep disorders, other).

The obesity of the patients was diagnosed according to the ICD-9 codes as follows: overweight, obesity, and other nutritional hypertrophy (ICD-9-CM code 278); overweight and obesity (ICD-9 CM code 278.0); morbid obesity (ICD)-9-CM code 278.01); overweight (ICD-9-CM code 278.02); and obesity hypoventilation syndrome (ICD-9-CM code 278.03).

The comorbidities evaluated in this study were DM (ICD-9-CM code 250), hypertension (ICD-9-CM code 401–405), hyperlipidemia (ICD-9-CM code 272.4), coronary heart disease (ICD-9-CM code 414.01), stroke (ICD-9-CM code 430–438), chronic heart failure (ICD-9-CM code 428.0), chronic obstructive pulmonary disease (ICD-9-CM code 490–496), chronic kidney disease (ICD-9-CM code 585), liver cirrhosis (ICD-9-CM code 571.5), tumor (ICD-9-CM code 199), anxiety (ICD-9-CM code 300.00), and depression symptoms (ICD-9-CM code 296.2–296.3, 300.4 and 311).

### 2.4. Statistical Analysis

The descriptive statistics were expressed in the form of percentages, averages, and standard deviations. The chi-square test and *t*-test were used to evaluate the distribution of categorical and continuous variables between the case and control groups. After adjusting for age, gender, education level, insurance premiums, comorbidities, Charlson’s comorbidity index (CCI), season, place of residence, level of urbanization, and level of care, a conditional logistic regression analysis was performed to evaluate the CSA and the impact of obesity risk. Conditional logistic regression was used to analyze the effects of the first and last CSA exposure on obesity factors before obesity diagnosis. According to the central limit theorem, (a) if the sample data are approximately normal then the sampling distribution too will be normal; (b) in large samples (>30 or 40), the sampling distribution tends to be normal, regardless of the shape of the data; and (c) the means of the random samples from any distribution will themselves have normal distribution [25]. As the dependent variable, obesity, is a categorical variable (obese and non-obese), logistic regression requires the dependent variable to be a continuous variable before it needs to be tested for normality. All analyses were performed using SPSS version 22 (IBM, Armonk, NY, USA). A *p*-value < 0.05 is considered to be statistically significant.

## 3. Results

The demographic data are shown in Table 1. The average age of the 121,815 patients was 44.30 ± 15.64 years, of which 42.77% were males and 57.23% were females. We screened 24,363 obese patients (cases) and 97,452 non-obese patients (control group). The prevalence of comorbidities in the case group was higher than in the control group. In the case group, the CCI, season, location of residence, level of urbanization, and level of care are significant.

P: Chi-square/Fisher exact test for categorical variables, and *t*-test for continuous variables. DM, diabetes; HTN, hypertension, CHD, coronary heart disease, CHF, chronic heart failure, COPD, chronic obstructive pulmonary disease, CKD, chronic kidney disease.

The logistic regression of the obesity variables is shown in Table 2. The risk of obesity in the CSA group was significantly greater than in the non-CSA group (AOR = 2.234, 95% CI = 1.483–4.380). In addition, patients aged 45–64 years and over 65 have a significantly lower risk of obesity than those aged 20–44 years (AOR = 0.608, 95% CI = 0.578–0.627, AOR = 0.422, 95% CI = 0.397–0.446). On the CCI score, every increase of 1 point in the CCI score of CSA patients increases the risk of obesity by 13.8%. In summer and winter, the risk of obesity was significantly reduced (AOR = 0.842, 95% CI = 0.804–0.885; AOR = 0.876, 95% CI = 0.834–0.921).

Using logistic regression to stratify the obesity factors of the listed variables, as shown in Table 3, the obesity risk of CSA patients is 2.234 times that of the control group (AOR = 2.234, 95% CI =1.483–4.380). The obesity risk of female CSA patients was 2.252 times that of the control group (AOR = 2.252, 95% CI =1.496–4.417). The risk of CSA obesity in patients aged 20–44 years was significantly higher than in the control group (AOR = 2.577, 95% CI = 1.713–5.054). Conditional logistic regression analysis showed that the risk of obesity diagnosis in the CSA group in spring (AOR = 2.547, 95% CI = 1.682–4.986) was significantly greater than in the control group.

Logistic regression analysis of obesity factors between exposures to CSA in different periods is shown in Figure 2. When compared with non-obese patients, obese patients are more likely to experience CSA (AOR = 2.234, 95% CI = 1.483–4.380). In addition, the closer the exposure time is to the obesity diagnosis time, the more the obesity severity shows a dose-response effect (CSA exposure < 1 year, AOR = 2.484, 95% CI = 1.689–4.672; CSA exposure ≥ 1 year, <5 years, AOR = 2.105, 95% CI = 1.417–4.028; CSA exposure ≥ 5 years, AOR = 1.862, 95% CI = 1.206–3.872).

Figure 3 shows that the average CSA exposure duration of obese patients is 2.234 times that of non-obese patients (AOR = 2.234, 95% CI = 1.483–4.380). The longer the exposure time, the more severe the obesity and the dose–response effect (CSA exposure < 1 year, AOR = 2.101, 95% CI = 1.312–3.879; CSA exposure ≥ 1 year, <5 years, AOR = 2.207, 95 %CI = 1.435–4.288; CSA exposure ≥ 5 years, AOR = 2.976, 95% CI = 1.589–5.120).

## 4. Discussion

The results of this study show that the risk of obesity in patients aged 45–64 years or over 65 is significantly lower than in patients aged 20–44 years. This may be due to weight loss or weight control in middle-aged and elderly people caused by the disease itself [26]. However, unknown factors may affect this result [18]. In addition, the risk of obesity in summer and winter is significantly lower than in spring, which is consistent with the results of Ma et al. [27]. The total daily intake in spring is higher than in autumn. The difference in the total daily intake is 222 kcal/day [27]. Lin et al. had similar results [28]. Although the physiological mechanism of the association between CSA and obesity is unclear, we infer the underlying mechanism from previous studies, which may provide some insights for our observations. The current research literature (mainly cross-sectional and observational studies) has not yet clarified whether sleep disorders cause obesity or obesity causes sleep disorders. More research is needed, including larger sample sizes and control of confounding factors [29].

CSA encompasses a variety of breathing patterns and clinical entities. They can be classified into two categories based on the partial pressure of carbon dioxide in the arterial blood. Non-hypercapnic CSA is usually characterized by a periodic breathing pattern, while hypercapnic CSA is based on hypoventilation. The latter form of CSA is associated with disorders of the central nervous system, neuromuscular, and rib cage, as well as obesity and certain medications or substance intake [30].

Compared with OSA in clinical practice, the relationship between obesity and CSA has received less attention. The clinical feature of CSA is the repeated apnea during sleep due to temporary sleep apnea [31]. CSA represents a disorder of several breathing patterns, not caused by a single breathing pattern disorder [31]. Verhulst et al. found evidence that obesity and adipose tissue are associated with CSA, reflecting the unstable breathing pattern of CSA [32]. Verhulst et al. believe that several hypotheses can explain the interaction between obesity and CSA: obesity leads to a decrease in the volume of the chest cavity resulting in a decrease in oxygen reserves, impaired response to hypoxia and hyperventilation, and too low ventilation due to leptin resistance leading to CSA; followed by obesity, fat reduction, or folding of the upper airway. More research is needed to clarify this relationship [33,34,35,36,37]. CSA occurs frequently in patients with heart failure (HF) and is caused by the alternation of hyperventilation and compensatory apnea in abnormal respiratory control [38,39]. In various case reports, the frequency of CSA in HF patients exceeds OSA, ranging from 21% to 40% [40,41,42]. CSA is associated with increased mortality [43]. CSA patients are different from OSA patients. Patients with HF and CSA do not have upper airway collapse. CSA may be regarded as the result of HF, which seems to be related to the severity of the disease [44,45]. It is directly proportional to the hemodynamic severity of HF, and nanostimulating peptides (NPs) also increase, and the increase in concentration is related to the presence of CSA [46,47].

Leptin is an auxiliary agent that regulates food intake and energy expenditure [48]. Since its discovery, leptin has become known as a pectoral muscle hormone that is widely studied in the context of cardiovascular disease and is important in the supervision of ventilation control [49,50,51]. Human research on leptin and ventilation is mainly carried out in patients with OSA or obesity and low-ventilation syndrome [52,53]. CSA is also associated with increased chemical sensitivity to CO_2_ [54] and excessive ventilation during rest and exercise [55,56]. In vitro studies have shown that peritoneal nutrient peptide (ANP) inhibits the secretion of leptin in adipocytes. This may be mediated by the increase of guanosine monophosphate in the loop, thereby initiating lipid dissolution [57,58]. In addition, leptin-deficient hearing/culling mice showed an increase in ventilation drive [59]. Adults with higher BMI tend to have higher serum leptin levels [60,61,62,63,64]. Leptin can directly act on the respiratory control center to increase the sensitivity of CO_2_ during sleep and subsequently prevent respiratory depression in obese subjects [65]. Therefore, the positive effect of leptin on ventilation can compensate for the increase in mechanical load and breathing disorders in obese individuals [56]. In contrast, one reason for obese hypoventilation in patients with CSA is the development of leptin resistance [66,67]. According to previous studies, the prevalence of obesity hypoventilation in adults with CSA is 13.8–20% [68,69,70].

In animals lacking leptin for hyperventilation, the expression of leptin in adipocytes is reduced by intracardiac nutrient peptides (ANP), and the increase of cyclic urinary peptides (NPs) is associated with an increased risk of CSA [71]. The concentration of leptin is inversely proportional to AHI, indicating that there is a significant relationship between the severity of CSA and the concentration of leptin. Logistic regression analysis showed that the concentrations of leptin and BNP are closely related to the presence of CSA in high-frequency subjects [71]. Obesity may still aggravate CSA through several mechanisms, such as stimulating upper airway mechanical senses to inhibit the respiratory center in the pharynx or reducing the volume of the chest cavity to reduce oxygen storage [72,73]. Verhulst et al. proved the ability to accurately predict CSA through the BMI z-score and waist circumference [32]. A higher degree of obesity may lead to increased brain displacement of the diaphragm, thereby reducing breathing power. Therefore, in obesity patients, the protective effect of leptin and the negative impact of fat distribution on central obesity will affect the outcome of the development of CSA [31].

The integrated analysis study by Zhou et al. (2019) found there was a reverse J-shaped relationship between sleep time and obesity; with 7–8 h sleep per day, the risk of obesity is the lowest [74]. Compared with seven hours of sleep per day, individuals who sleep less than seven hours per day have a relative risk of obesity of 1.09 (95% CI 1.05–1.14) for each hour less than seven hours; individuals with longer sleep durations have a relative risk of 1.02 (95% CI 0.99–1.05) for each one-hour increment [74]. Short sleep time can significantly increase the risk of obesity. Compared with seven hours of sleep per day, with less than seven hours of sleep, each hour of decrease in sleep duration increases the risk of obesity by 9% [65]. Whether short-term or long-term sleep is a risk factor for obesity is the focus of debate. Two integrated analyses [75,76] explored the relationship between sleep time and the risk of obesity in adults, showing that the length of sleep has a significant relationship with the increased risk of obesity. Zhou et al. (2019) used dose-response analysis to analyze sleep, and the relationship between time and obesity risk was quantitatively assessed [74]. The research of Zhou et al. (2019) also includes more comprehensive original documents and has more rigorous methodological power to improve the accuracy and reliability of the research results [74]. In summary, the above studies seem to indicate a U-shaped correlation between sleep duration and weight gain [77]. Meanwhile, our research results show that the risk of non-obesity for CSA is 2.234 times that of patients with non-CSA. The closer the exposure time for CSA is to the obesity diagnosis time, the more serious the obesity is, and there is a dose–response effect. The risk of obesity in patients with CSA for the duration of exposure is 2.234 times that of patients with non-CSA. The longer the exposure time, the more severe the obesity; there is also a dose-response effect.

Furthermore, it is worth noting that the genes expressed in the brain at key BMI-related sites, such as FTO, MC4R, MAP2K5, GNPDA2, PCSK1, and BDNF, may have connections with sleep time [78]. Previous studies have shown that shortened sleep time (sleep time < 7 h/day) increases the risk of genetic expression of high BMI (average 36.6 years) in American adults [78], resulting in different genetic effects on BMI depending on sleep times [79]. However, the mechanism by which lack of sleep increases the genetic risk on the pathogenesis of obesity is less explored. At present, adipose tissue has been identified as a dynamic endocrine organ that produces a series of biologically active substances, collectively referred to as “adipose”, which regulates energy balance, inflammation, insulin resistance, and cardiovascular function [80]. Among the six genes related to leptin, three genes (MC4R, BDNF, and PCSK1) are known to be related to the hypothalamic leptin-melanin pathway [81,82,83].

The above pathophysiological mechanism can explain the relationship between CSA and obesity in this study. Our results show that the prevalence of CSA in obese patients is 2.34 times that of non-obese patients. In addition, the duration of CSA and the closeness to the study time is positively correlated with the severity of obesity. Therefore, it is necessary to consider the relationship between the occurrence and duration of CSA and obesity.

This study has several limitations. First, the NHIRD did not provide detailed information that might affect our survey results, such as information related to drinking, smoking, eating, and physical exercise behavior. Second, although this study was carefully designed and controlled for confounding factors, bias may still exist due to unmeasured or unknown confounding factors—for example, the onset of depression, the stage of obesity at the time of diagnosis, and drugs that may affect the outcome. Finally, BMI was not among the variables in our study. It is recommended that a prospective cohort study be conducted to evaluate the relationship between CSA and obesity.

## 5. Conclusions

The study showed that obese patients experience more severe CSA than non-obese patients do. Further, the proximity of CSA to the study time and the duration of exposure are positively correlated to the severity of obesity and have a dose-response effect. CSA may be a risk factor for obesity. Healthcare providers should pay close attention to the relationship between CSA and obesity risk.

## Figures and Tables

**Figure 1 ijerph-19-01289-f001:**
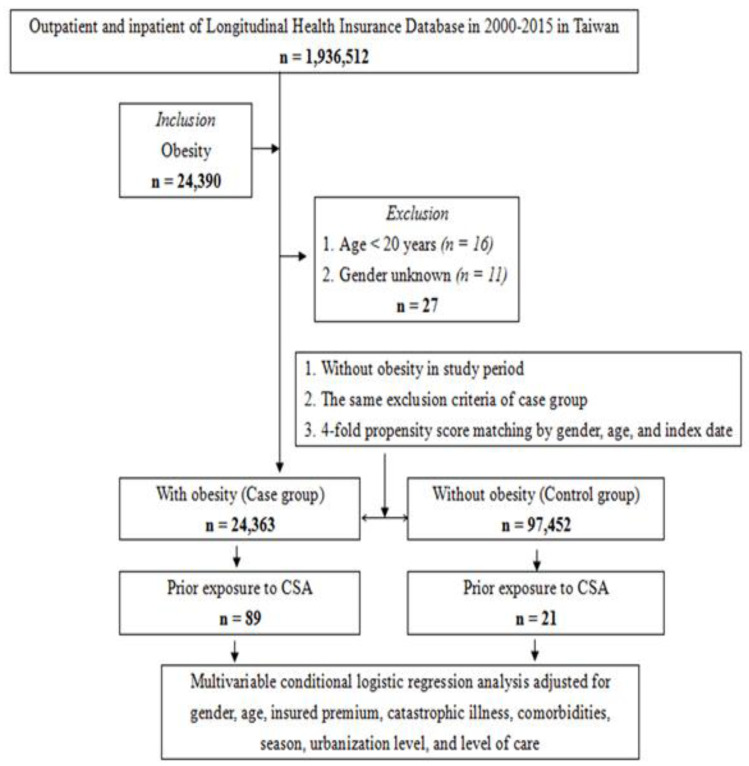
The flowchart of the study design from National Health Insurance Research Database in Taiwan.

**Figure 2 ijerph-19-01289-f002:**
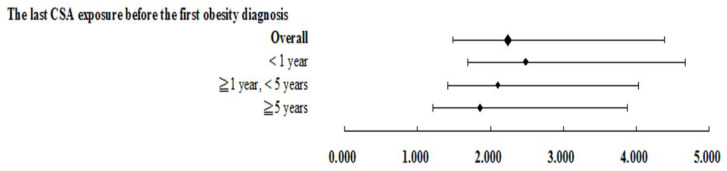
The last CSA exposure before the first obesity diagnosis.

**Figure 3 ijerph-19-01289-f003:**
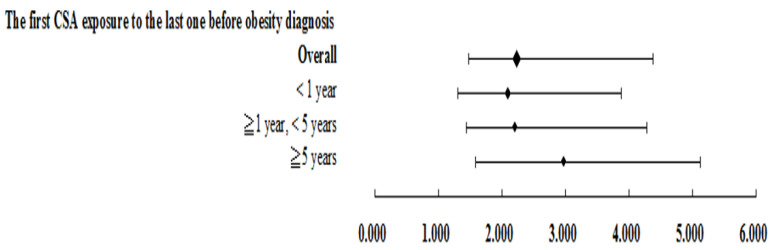
The first CSA exposure to the last one before obesity diagnosis.

**Table 1 ijerph-19-01289-t001:** Characteristics of the patients.

Obesity	Total		With		Without		*p*-Value
Variables	n	%	n	%	n	%	
Total	121,815		24,363	20.00	97,452	80.00	
**CSA**							<0.001
Without	121,705	99.91	24,274	99.63	97,431	99.98	
With	110	0.09	89	0.37	21	0.02	
**Gender**							0.999
Male	52,105	42.77	10,421	42.77	41,684	42.77	
Female	69,710	57.23	13,942	57.23	55,768	57.23	
Age (years)	44.30 ± 15.64		44.25 ± 15.53		44.31 ± 15.67		0.592
**Age Group (Years)**							0.999
20–44	74,135	47.48	14,827	47.48	59,308	47.48	
45–64	34,330	21.99	6,866	21.99	27,464	21.99	
≥65	47,680	30.54	9,536	30.54	38,144	30.54	
CCI_R	0.05 ± 0.27		0.06 ± 0.35		0.05 ± 0.24		<0.001

**Table 2 ijerph-19-01289-t002:** Logistic regression of obesity variables.

Variables	Adjusted OR	95% CI	*p*-Value
**CSA**			
Without	Reference		
With	2.234	1.483–4.380	<0.001
**Gender**			
Male	0.862	0.645–1.089	0.171
Female	Reference		
**Age group (Years)**			
20–44	Reference		
45–64	0.608	0.578–0.627	<0.001
≥65	0.422	0.397–0.446	<0.001
CCI_R	1.138	1.093–1.386	<0.001
**Season**			
Spring	Reference		
Summer	0.842	0.804–0.885	<0.001
Autumn	1.007	0.962–1.057	0.863
Winter	0.876	0.834–0.921	<0.001

OR = odds ratio, CI = confidence interval, adjusted OR = adjusted variables listed in the table.

**Table 3 ijerph-19-01289-t003:** Factors of obesity stratified by variables using logistic regression.

Group	With CSA vs. Without CSA (Reference)		
Stratified	Adjusted OR	95% CI	*p*-Value
Overall	2.234	1.483–4.380	<0.001
**Gender**			
Male	2.213	1.438–4.432	<0.001
Female	2.252	1.496–4.417	<0.001
**Age Group (Years)**			
20–44	2.577	1.713–5.054	<0.001
45–64	2.256	1.501–4.436	<0.001
≥65	1.581	1.050–3.102	0.001
**Season**			
Spring	2.547	1.682–4.986	<0.001
Summer	1.952	1.288–3.835	<0.001
Autumn	2.556	1.693–5.000	<0.001
Winter	2.082	1.381–4.214	<0.001

Adjusted OR = adjusted odds ratio: Adjusted for the variables listed in Table 3; CI = confidence interval.

## Data Availability

Data are available from the NHIRD published by the Taiwan NHI administration. Because of legal restrictions imposed by the government of Taiwan concerning the “Personal Information Protection Act,” data cannot be made publicly available. Requests for data can be sent as a formal proposal to the NHIRD (http://www.mohw.gov.tw/cht/DOS/DM1.aspx?f_list_no=812 accessed on 15 October 2021).

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
