# Peer review of "Obese Patients Experience More Severe CSA than Non-Obese Patients"

_ijerph, 2022, doi:10.3390/ijerph19031289_

Round 1

Reviewer 1 Report

Ching Huang et al. performed an interesting analysis regarding the association between CSA and obesity.

Introduction section should be shorter and more focused in introducing the reader to the actual aim of the study (e.g. remove lines 47-51).

Statistical analysis should be defined in more details. Was normality tested? 

The biggest setback of this study is lack of BMI as a variable. In this regard, the authors failed to address studies in which BMI was not shown to be a significant predictor of CSA (actually the authors did not address any study exploring this issue).

The authors should present their main result as a form of a figure.

Reviewer 2 Report

Introduction

- Certainly the authors stressed the concept of obstructive sleep apnea but not correctly discussed the link between obstructive OSA in obese patients, especially in children. Please discuss and cite doi: 10.1002/ppul.23639.  and doi: 10.3390/children8100921.

  • line 73, as reported sleep fragmentation and chronic inflammation status could lead to neurocognitive disorders, with poor cognitive functions and memory up to degenerative disease as Alzheimer of Parkinson. Please discuss and cite doi: 10.3390/bs11120180. and doi: 10.3233/JAD-179936.

Methods

  • a flow diagram of the study protocol was uploaded?
  • which guidelines where adopted for the study?
  • define better the criteria for obesity

Results

clearly described. Good work

Discussion

  • please always use could, lead not imperative when state a concept
  • Central sleep apnea (CSA) encompasses a variety of breathing patterns and clinical entities. They can be classified into 2 categories based on the partial pressure of carbon dioxide in arterial blood. Non-hypercapnic CSA is usually characterized by a periodic breathing pattern, while hypercapnic CSA is based on hypoventilation. The latter form of CSA is associated with disorders of the central nervous system, neuromuscular and rib cage, as well as obesity and certain medications or substance intake. In contrast, non-hypercapnic CSA is typically accompanied by over-ventilation and often associated with heart failure, cerebrovascular disease, and staying at high altitudes. CSA and hypoventilation syndromes are often considered separately, but the pathophysiological aspects often overlap. An integrative approach helps to recognize the underlying pathophysiological mechanisms and to choose suitable therapeutic strategies. Therefore it is necessary to raise awareness for the different pathophysiological components and discuss the evidence in order to enable targeted therapeutic strategies.please discuss and cite doi: 10.1159/000500728.

Round 2

Reviewer 1 Report

No further comments.